# Comparative Ungual Drug Uptake Studies: Equine Hoof Membrane vs. Human Nail Plate

**DOI:** 10.3390/pharmaceutics14122552

**Published:** 2022-11-22

**Authors:** Dorota Dobler, Mona Gerber, Thomas M. Schmidts, Frank Runkel, Peggy Schlupp

**Affiliations:** 1Technische Hochschule Mittelhessen, Institute of Bioprocess Engineering and Pharmaceutical Technology, 35390 Giessen, Germany; 2Faculty of Biology and Chemistry, Justus Liebig University, Ludwigstraße 23, 35390 Giessen, Germany; 3Institute of Pharmaceutics and Biopharmaceutics, Philipps University, Robert-Koch-Straße 4, 35037 Marburg, Germany

**Keywords:** caffeine, testosterone, sorbic acid, human nail plate, ungual permeation, ungual penetration, permeation device, equine hoof membrane

## Abstract

Human nail diseases, mostly caused by fungal infections, are common and difficult to treat. The development and testing of new drugs and drug delivery systems for the treatment of nail diseases is often limited by the lack of human nail material for permeation studies. Animal material is frequently used, but there are only few comparative data on the human nail plate, and there is neither a standardized test design nor a nail bed analogue to study drug uptake into the nail. In this study, a new permeation device was developed for permeation studies, and the permeation behavior of three model substances on the human nail plate and a model membrane from the horse hoof was investigated. A linear correlation was found between drug uptake by the human nail plate and the uptake by the equine hoof. The developed and established permeation device is suitable for investigations of ungual drug transport and enables the use of different membrane diameters and the use of a gel-based nail bed analog. The hydrogel-based acceptor medium used ensures adequate stabilization and hydration of the nail membrane.

## 1. Introduction

The human nail plate is mainly composed of sulfur-rich α-keratin (~80%) and, in contrast to the skin, exhibits a higher water and a lower lipid content (0.1–1.0% lipids) [1]. It is generally considered to be composed of three layers, called the dorsal, the intermediate, and the ventral nail plate. The tissue under the nail plate, the nail bed, consists of a viable epidermis. Each of the layers has unique physical properties, and thus a special drug permeability through each layer [2].

Bacterial or fungal infections (about 50%), diseases due to inflammatory or metabolic conditions (15%), and malignancies and pigment disturbances (5%) represent the most prevalent nail disorders [3]. Common nail diseases such as onychomycosis and psoriasis require systematic long-term therapies [4]. An oral treatment for these diseases has several safety limitations because of systemic drug side effects and drug–drug interactions. Moreover, due to the limited vascular access to the nail plate and its barrier properties, a systemic treatment of nail diseases is often ineffective. Thus, a local treatment is the preferred medical choice. However, the challenge in ungual drug delivery is the effective barrier of the nail plate, as this barrier can reduce or even prevent adequate amounts of drug absorption in the target tissue.

Drug permeation across the nail plate is influenced by the physicochemical properties of the active pharmaceutical ingredient (API), by interactions between the API and the keratin structure, as well as by the chosen drug carrier system. To develop an effective drug delivery system, a suitable in vitro nail plate model is essential. The use of native nail material would be the first choice but is usually not feasible due to a lack of availability. Different nail permeation models, e.g., keratin films made from human hair [5] or polymeric membranes [6], have been proposed in the last decade. However, these model membranes are cost-intensive, their production is time-consuming, or there are still large differences in the composition and permeation behavior of the polymeric model membranes.

Mertin and Lippold [7] reported that the human nail plate and the keratin membrane from bovine hooves have similar properties and behave like hydrophilic gel membranes. Both human nails and animal hooves are composed of the same keratin type (i.e., α-keratin). However, bovine hooves have been reported to be more permeable than the human nail plate, due to their less dense keratin matrix [8]. Further, the hoof proteins have significantly less cystine residues, and most likely fewer disulfide links compared to the human nail plate [9]. As a result, the permeability through the hoof membrane, in comparison to the human nail plate, could be less influenced by penetration enhancers that break the disulfide linkages. Nevertheless, bovine hooves have regularly been used as an alternative to the human nail in permeation studies [10,11,12]. In this context, only limited consideration is given to the frequently used aqueous acceptor medium, which leads to a strong hydration of the material, and thus has a decisive influence on permeation.

However, to date, no commonly accepted in vitro nail model has been developed for the testing of new drugs and formulations for an ungual application. Furthermore, the current data are not sufficient to correlate animal hooves and the human nail plate, and thus use this material as a model for the human nail plate in permeation studies. Studies with active ingredients that differ in their physicochemical properties are required to determine the functional suitability of the equine membrane.

In this work, a novel permeation device was designed that enables the effective use of small nail samples. Moreover, a carbomer gel was developed as a nail bed analogue, which guarantees a sufficient stabilization and hydration of the nail membrane. The investigated drugs are recommended by the OECD, the water-soluble caffeine and poorly soluble testosterone. Additionally, sorbic acid, a pH-dependent model substance, was used [13]. The equine hoof membrane was used as the model membrane.

## 2. Materials and Methods

### 2.1. Preparation of the Equine Hoof Plate and the Human Nail Plate

Equine hooves were obtained directly from a local slaughterhouse. First, the hooves were roughly cleaned with ethanol, wiped with paper towels, and stored in the freezer (−18 °C) until further processing. The front and the side plates of the horse hoof were removed by cutting it into three pieces using a common jig saw. These pieces were used to prepare nail membranes out of the equine hoof with a band saw. To ensure the comparability to human nail material, the thickness of the hoof membranes was set at 400 ± 100 µm. Obtained hoof membranes were hydrated for at least 30 min between two water-soaked cloths for punching out 6 mm membranes for later use in the permeation device. These membrane discs were weighed with a metal plate during drying at room temperature for 24 h to prevent membrane formation. Only nails without macroscopic defects were used for the examinations. Hoof membrane discs were stored at room temperature.

Human nail material was collected from healthy female volunteers aged between 35 and 60 years, who allowed their nails to grow appropriately. The nail material was used without any further pre-treatment. The nails were free from cracks and nail polish. They were hydrated as described above and punched out with a 6 mm hollow punch.

### 2.2. Preparation of the Nail Bed Analogue

To simulate the nail bed during the permeation studies, a high cross-linked acrylic acid polymer (Tego**^®^** Carbomer 140, Evonik Industries AG, Essen, Germany) was used as the acceptor phase. For the permeation experiments with caffeine and sorbic acid, 0.5% Carbomer 140 was dissolved in water by magnetic stirring. For the experiments with testosterone, 0.5% Igepal**^®^** CA-630 (Merck KGaA, Darmstadt, Germany) was added additionally. Sol-gel transformation was induced by increasing the pH value up to about 6.2 by the application of a 10 M sodium hydroxide solution.

### 2.3. Characterization of the Nail and Hoof Samples

The total water absorption capacity of the human nail and equine hoof material was determined by incubating the prepared membrane discs in water for 120 h. Samples were then removed, excess moisture wiped off, and the sample weights determined (*n* = 3).

To confirm the use of the nail bed analogue, the water absorption of the hoof membrane was also compared in the presence of water as well as the nail bed analogue. For this, samples were dried overnight, weighed, and then covered with 1 mL of water; or the nail bed analogue was incubated for 120 h at room temperature in a sealed container. At defined time points, the samples were removed from the containers, excess moisture was wiped off, and the sample weights were determined (*n* = 3).

### 2.4. Establishment of the Novel Permeation Device

To study the penetration and permeation of the three model substances in and across the equine hoof and the human nail plate, respectively, a two-chamber system was developed (Figure 1). The permeation devices were made of stainless steel (1.4305) and met the requirements for a reduced permeation area, especially for human nail plates and the use of the nail bed analogue. The diffusion area of the permeation device was 0.785 cm^2^ (acceptor volume of 1.218 mL), respectively 0.126 cm^2^ (acceptor volume of 1.020 mL) using a Teflon adapter. The reduced diffusion area set-up was used for the following permeations studies.

### 2.5. Drug Penetration and Permeation Studies

Penetration and permeation studies of caffeine, sorbic acid, and testosterone were performed using the described modified vertical diffusion cell (Figure 1). Human nail plates and equine hoof plates were pre-hydrated between two water-soaked cloths for 30 min, respectively. The membranes were clamped between the two seals and tested for integrity by a leak test using a rubber seal and a syringe. The nail bed analogue was filled into the acceptor compartment while avoiding air bubbles to ensure the contact between the membrane and the gel. A total of 100 µL of the aqueous donor solution, caffeine, sorbic acid (both in two concentrations: 1 mg/mL and 200 µg/mL), or testosterone (200 µg/mL with 0.4% ethanol and 2% Igepal**^®^** due to a low aqueous solubility) was applied onto the membrane, in sufficient quantity to cover the equine hoof and the human nail plate, respectively. The sealed permeation devices were incubated at 32.5 °C for 24 h and 120 h, respectively. At the end of incubation, the remaining donor solutions (three rinsing steps), the nail and hoof membranes, and the acceptor medium were analyzed by HPLC. The acceptor medium and gel residues on the bottom of the membranes were collected by three washing steps with water (caffeine and sorbic acid) or 2% Igepal**^®^** aqueous solution (testosterone). The gel was liquefied by adding 80 µL of 6 M hydrochloric acid and by intensive mixing. The resulting suspension was centrifuged and the supernatants were analyzed. The drug recovery out of the membranes was conducted by three extraction steps (1 h, 2 h, overnight) at 50 °C. Thereafter, the nail and hoof membranes were cut in pieces and were transferred into 2 mL vials containing the extraction agents. The corresponding mobile phases (caffeine: 10% acetonitrile (ACN) and 90% phosphate buffer; sorbic acid: 30% ACN and 70% phosphate buffer, and testosterone: 45% ACN and 55% phosphate buffer) were used as extracting agents and the supernatants of each extraction step were analyzed.

### 2.6. Stability of Sorbic Acid in the Presence of Keratin

A total of 30 mL of sorbic acid solution in distilled water (50 µg/mL) was incubated at 32.5 °C with 300 mg of hydrolyzed keratin powder (Crotein K-PW-(WD), Croda, Snaith, UK) to study the stability over time by HPLC.

### 2.7. Quantification of Caffeine, Sorbic Acid, and Testosterone

The amount of caffeine, sorbic acid, and testosterone (all received by Caesar & Loretz GmbH, Hilden, Germany) was analyzed by HPLC analysis, as described by Schlupp et al. [13]. In detail, caffeine, sorbic acid (Caesar & Loretz GmbH, Hilden, Germany, both), and testosterone (Sigma Aldrich, Darmstadt, Germany) were quantified by HPLC (LaChrom Elite**^®^** HPLC system, VWR International GmbH, Darmstadt, Germany) and UV detection at 230 nm (caffeine), 255 nm (sorbic acid), or 245 nm (testosterone), respectively. A LiChrospher**^®^** 100 RP-18e (5 µm) LiChroCART**^®^** 125-4 column (Merck KGaA, Darmstadt, Germany) was used for all test substances. The isocratic mobile phase was 10% acetonitrile (ACN) and 90% phosphate buffer (10 mM, pH 2.6: 0.34 mL/L orthophosphoric acid (85%) and 0.68 g/L NaH_2_PO_4_·H_2_O), delivered with a flow of 1.0 mL/min for caffeine (retention time = 3.2 min, LOD: 5 ng/mL and LOQ: 14 ng/mL), and 30% ACN and 70% phosphate buffer with a flow of 1.2 mL/min for sorbic acid (retention time = 2.3 min, LOD: 9 ng/mL and LOQ: 26 ng/mL); the column temperature was 40 °C for each. For testosterone, the mobile phase was a gradient of ACN/phosphate buffer (45:55–85:15 *v*/*v* within 10 min followed by a washing procedure) with a flow of 1.0 mL/min. The retention time was 5.1 min, LOD: 23 ng/mL and LOQ: 69 ng/mL; the column temperature was 40 °C.

### 2.8. Data Analysis

The results of the penetration and permeation studies were presented in Box-and-Whisker plots (min. to max.): The whiskers extend from the smallest value to the largest. The top and bottom of the box are the 25th and 75th percentiles and the median within. All experiments were performed 6 times. Pearson correlation coefficient between the drug uptake by the human nail plate and uptake by the hoof membrane was determined. Images and correlations were generated using GraphPad Prism**^®^** software, version 8.03.

## 3. Results

### 3.1. Characterization of the Nail and Hoof Samples

The water absorption capacity of the equine hoof membranes in comparison to the nail material was determined by an incubation of the samples in water for 120 h. The total water uptake was 31.8 ± 6.4% for hoof membranes and 18.57 ± 1.38% for the human nail.

To determine the impact of the acceptor medium on the hydration of the membrane, the hoof membrane was incubated in water or gel (nail bed analogue). A rapid increase in the water uptake in both approaches, i.e., water (32.0 ± 8.1%) and gel (26.5 ± 4.7%), within the first 5 h took place (Figure 2). Subsequently, an equilibrium was reached for both samples. The total water uptake was 33.5 ± 7.2% (water) and 29.3 ± 3.3% (gel) after 120 h of incubation.

### 3.2. Penetration and Permeation Studies of Caffeine, Testosterone, and Sorbic Acid

Drug uptake (penetration + permeation) studies were performed using human nails and hoof membranes over 24 h (20 µg, Figure 3 and 100 µg, Figure 4) and 120 h (20 µg, Figure 5). Regardless of the membrane used, the average overall recoveries of caffeine and testosterone were 101.1 ± 4.8 % and 103.6 ± 1.69%, respectively. The recovery of sorbic acid was lower by comparison, and a dependence on the incubation time was observed (Table 1). In addition, the recovery of sorbic acid, using the hoof membrane, was lower as compared to the human nail.

Due to the lower recovery of sorbic acid, depending on the biological material used, the experimental time, and the observed changes in the HPLC spectrum compared to the standard solution, further experiments were performed. Here, sorbic acid was incubated with keratin powder, and the sorbic acid content was determined over time by an HPLC. A continuous decrease in the sorbic acid concentration (Appendix A) and the simultaneous appearance of additional peaks in the chromatogram (Appendix A) were observed. This confirms the hypothesis that keratin was responsible for the loss of sorbic acid.

In general, we found that the drug uptake in the membrane and acceptor medium was higher in the hoof membrane than in the human nail (Figure 3, Figure 4 and Figure 5), regardless of the study duration, the drug, and the drug concentration.

A linear correlation was found between the penetrated and the permeated amount of the drug in the human nail and that in the equine hoof membrane, respectively (Figure 6). The results are described with the following equations: $drug_{nail}\left[µg\right]=\frac{drug_{hoof}\;\left[µg\right]-0.324}{1.541}$ for the penetrated drug,$\;drug_{nail}\left[µg\right]=\frac{drug_{hoof}\;\left[µg\right]-0.499}{1.480}$ for the permeated drug and $drug_{nail}\left[µg\right]=\frac{{\mathrm{drug}}_{hoof}\;\left[µg\right]-0.757}{1.544}$ for total drug uptake.

A time-dependent change of the drug concentration in the membrane was observed (Appendix A). The amount of caffeine and testosterone in the membrane was more than two times higher after 120 h than after 24 h, indicating that an equilibrium state was not reached after 24 h. In contrast, the amount of sorbic acid was about two times lower after 120 h than after 24 h.

## 4. Discussion

The topical treatment of nail diseases remains a challenge due to a limited drug absorption. To develop and evaluate novel drug delivery systems, in vivo conditions must be reproduced as closely as possible in the laboratory. Ungual absorption has been systematically studied in the last decades with different in vitro models. Models used for the human nail plate include bovine hoof membranes, nail sections from healthy human volunteers, and human cadaveric nail plates. However, the comparability of the different studies is difficult as the parameters are not standardized and the amount of comparative data is insufficient for the human nail plate. The use of human nail sections in the standard Franz diffusion cells is complicated due to their size. The generation of comparative data usually fails due to the absence of suitable test systems that allow experiments to be performed under conditions close to those in humans. Therefore, few comparable studies have been performed with human nail plates thus far.

For our studies, we have developed a permeation device that enables the comparison of the penetration and the permeation of the human nail plate and the hoof membrane, respectively, under identical conditions considering the small diffusion area. The diffusion area can be varied with an adapter and a hydrogel can be used as an acceptor medium. The carbomer gel mimics the condition of the nail bed better than a liquid solution. The acceptor medium plays an essential role in the membrane hydration and can affect its permeability significantly. A higher water content leads to a significantly higher flux due to the increased flexibility and segmental mobility of the keratin matrix [14]. As shown in the present experiments, the use of an aqueous solution leads to a considerably higher hydration of the membrane than with the carbomer gel. Thus, the carbomer gel corresponds more to the real conditions of a nail bed. A defined hydration of the nail plate during permeation tests is important, as an excessively high water content in the nail plate enhances penetration [14]. The acrylic acid polymer is characterized by its good stability at a neutral pH and tolerance at physicochemical sodium chloride concentrations (0.9% *w*/*w*). Furthermore, the pH-dependent sol-gel conversion allows a good preparation of the acceptor phase for analysis.

Reliable data interpretation in pharmacokinetic studies essentially requires the consideration of the substances’ stability under the experimental conditions. Caffeine, testosterone, and sorbic acid are stable in the donor and acceptor medium over the entire experimental period. However, a decrease in sorbic acid was observed in the presence of hoof and nail material. This may be due to interference between sorbic acid and the keratin protein, resulting in the formation of 5-(S-cysteinyl)-3-hexanoic acid due to the high content of cysteine in the human nail and in the hoof membrane [15]. A stronger decrease in sorbic acid, while using the hoof membrane despite a lower content of cysteine-containing proteins [16], could be related to an increased permeability, and consequently a higher reaction probability. Here, further investigations are necessary to confirm this hypothesis. The behavior of sorbic acid in the presence of hydrolyzed keratin cannot be directly compared with native keratin. During hydrolysis, disulfide and peptide bonds break, and the structure of keratin hydrolysates differs from the structure of keratin protein. Nonetheless, our experiment confirmed the occurrence of an interaction between sorbic acid and the amino acids of the hydrolyzed keratin.

In general, the transungual permeation is influenced by the physiochemical properties of the drug molecule, such as the molecular size, the molar volume, the partition coefficient (logP), and the charge [7,17]. Additionally, the properties of the drug formulation (e.g., the type of carrier system, the pH value, the drug concentration), the presence of penetration enhancers, nail properties (e.g., thickness, hydration), and the interactions between the permeating molecule and the keratin network of the nail plate, are also important [17].

The nail plate consists of many keratin strands, linked together by disulfide bonds. The penetration of molecules is limited by the spaces between the strands. For alkyl nicotinates, it has already been shown that the nail plate permeability coefficient decreased with increasing molecular size of the molecule [18]. The lipid content of the nail plate is up to 1%, while the water content varies between 10% and 25% and is directly related to the relative humidity [19]. In the presence of an aqueous medium or drug solution, the nail plate swells and the distance between the keratin fibers increases, creating a looser network that facilitates the permeation of the molecules, especially hydrophilic ones. The rather hydrophilic character of the nail plate is responsible for the fact that nail permeability increases with a decreasing partition coefficient logP [20]. However, the effect of the lipophilicity seems to be less pronounced than the effect of the molecular weight of the permeate [21], and thus can only be detected for substances with a similar molecular weight and marked differences in lipophilicity.

Our results confirm the correlation between molecular properties (Table 2) and nail permeability. The small and more hydrophilic caffeine shows a considerably higher degree of permeation as compared to testosterone. However, for lipophilic drugs such as testosterone the transition between the nail plate and hydrophilic acceptor medium seems to form an additional barrier. Therefore, the permeation of testosterone is very low in comparison to other tested drugs. The uptake of sorbic acid is considerably higher than the uptake of both other drugs, with the greater amount located in the membrane rather than in the acceptor medium (Figure 4). This effect could be due to three fundamental phenomena: the effect of permeant ionization on its solubility, the electrostatic interactions between the charges of the ionized permeant with the charges of the nail keratins, and the apparent increase in molecular weight due to ion hydration. Based on our experimental data, we suggest that the high content of sorbic acid in the membrane, compared to caffeine, is related to size and the keratin-binding capacity of sorbic acid, rather than the lipophilicity. This could be attributed to the ionic properties of sorbic acid, which may affect the uptake [22] and the distribution between the different compartments.

The net charge of the human nail is related to the proteins contained therein (the isoelectric point of acidic keratins in hoof and nail is between 4.7 and 5.6 [23]), depending on the pH value and, accordingly, the influence on the binding capacity for active ingredients. The pH of the donor solutions used is dependent on the sorbic acid concentration, 3.89 (200 µg/mL) and 3.35 (1 mg/mL), respectively; thus, in our experiments, the sorbic acid is mainly uncharged and negatively charged only to a small degree (<10%). In contrast, the pH value of the carbomer gel in the acceptor chamber is 6.2, which means that the sorbic acid will be almost completely dissociated and thus negatively charged. Consequently, the distribution of the sorbic acid between the acceptor medium and the nail plate seems to occur preferentially on the side of the nail plate.

Overall, our results for the three substances studied show that penetration and permeation are higher for hoof membranes than for human nails. Both nail and hoof are composed of horn-like keratins and differences between their membranes have been described in the literature. Hoof proteins have significantly fewer cystine residues and probably fewer disulfide bonds than the human nail [24,25]. The filaments are arranged differently: In the human nail, the filaments are perpendicular to the growth axis, whereas in the hooves and claws of most animals, they are orientated in the direction of growth [26]. The pore diameter of both membranes is described to be similar (approximately 10 μm on average), but hoof membranes generally have more pores per unit area (“more leaked”) [9]. In addition, the water absorption and the swelling behavior of hoof and nail are driven by their morphological differences [8]. The porous microstructure is modified by hydration, especially in the hoof [9]; accordingly, hooves are reported to be up to 30 times more permeable than the human nail plate [7]. The water absorption capacity of the nail is found to be about 25%, which corresponds to a doubling of its usual water content [27]. Using a water–ethanol mixture (80:20 *v*/*v*), the weight gain of human nails (about 27%) is much lower than that of hooves (about 40%) [8]. Our results support the previous findings on hydration in the literature. The hydration of the equine hoof in water is about 13% higher than in the human nail samples, which supports the assumption of a higher drug uptake in a hoof membrane.

To our knowledge, no valid comparative studies have yet been conducted with different animal hooves. However, there is some evidence that for drug uptake studies, the equine hoof may be more suitable as a model membrane for the human nail. Naumann et al. [28] reported a more similar penetration behavior between equine hooves and human nails, as compared to equine hooves and bovine hooves. Our results confirm the suitability of the hoof membrane as a nail model, independent of the drugs tested. The drug-independent correlation (equine hoof:human nail), with the factor of approximately 1.54 (R^2^ = 0.985) for penetration, 1.48 (R^2^ = 0.616) for permeation, and 1.54 (R^2^ = 0.986) for summary uptake, indicates that despite the differences between the materials used, similar underlying processes dominate the penetration and permeation behavior of the equine hoof and the human nail. The similar chemical composition of both membranes suggests similar interactions with drugs or other ingredients of drug delivery systems. The larger amount of equine hoof material available compared to the hoof material of other animals is another advantage.

## 5. Conclusions

In summary, it was shown that the equine hoof can be used as a nail model. The performed correlation between the equine hoof membrane and the human nails demonstrates the possibility of a prediction of ungual bioavailability obtained in studies on equine hooves. Since hoof membranes from horses are sufficiently available, they can be consistently used for the evaluation of new drug delivery systems and thus contributes to the improvement of ungual therapy. Further comparative studies are necessary in order to examine the extent to which the correlation is also transferable to the use of, for example, penetration enhancers. Otherwise, an adaptation is necessary. The greater number of disulfide bonds in nail keratin probably make nails more sensitive to agents that disrupt disulfide bounds [9].

In addition, we have developed and established a novel permeation device for samples with a small permeation area in the present study. The use of a hydrogel-based acceptor medium addresses the relevant hydration status of membranes in ungual uptake studies.

## Figures and Tables

**Figure 1 pharmaceutics-14-02552-f001:**
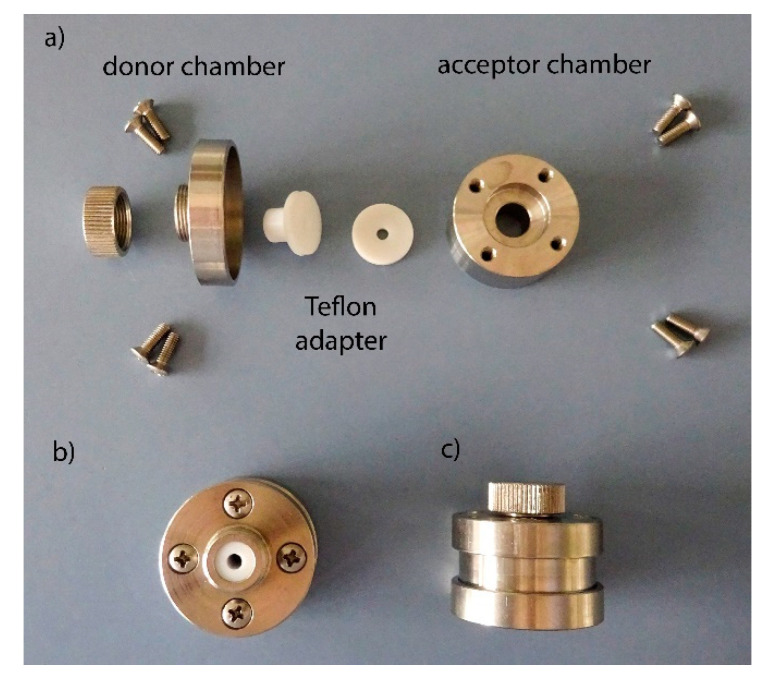
Two-chamber permeation device. Individual components (**a**), assembled permeation gadget top view with Teflon adapter (**b**), closed permeation device (**c**).

**Figure 2 pharmaceutics-14-02552-f002:**
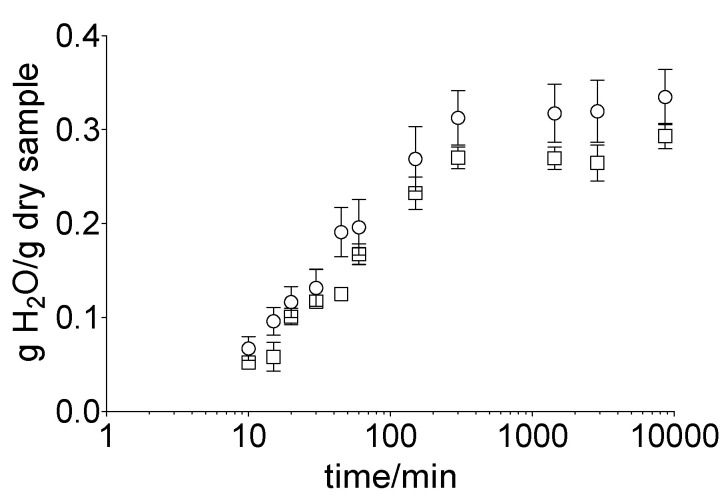
Water content of the hoof membrane over 120 h, depending on the incubation medium, water (ο) or gel (□), obtained by gravimetric measurements. Mean value ± standard deviation, *n* = 6.

**Figure 3 pharmaceutics-14-02552-f003:**
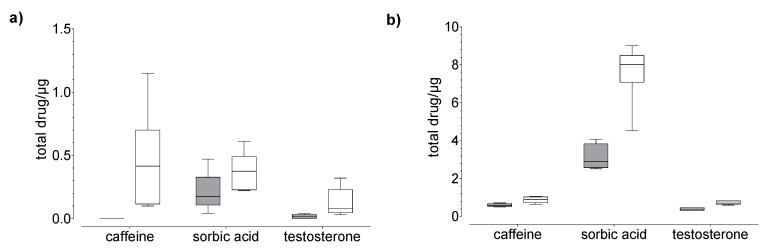
Results of the drug (**a**) permeation through and (**b**) penetration into nail (grey) and hoof (white) membrane after 24 h of exposure to caffeine, sorbic acid, and testosterone (donor: 20 µg). Box plot with median, minimum, maximum, and the 0.25 and 0.75 quartiles. *n* = 6.

**Figure 4 pharmaceutics-14-02552-f004:**
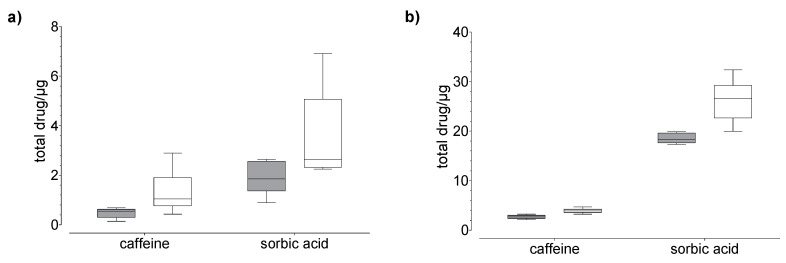
Results of the drug (**a**) permeation through and (**b**) penetration into nail (grey) and hoof (white) membrane after 24 h of exposure to caffeine and sorbic acid (donor: 100 µg). Box plot with median, minimum, maximum, and the 0.25 and 0.75 quartiles, *n* = 6.

**Figure 5 pharmaceutics-14-02552-f005:**
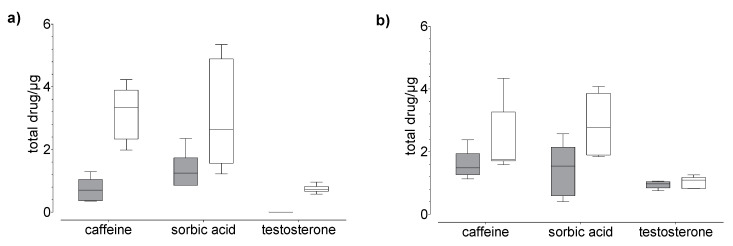
Results of the drug (**a**) permeation through and (**b**) penetration into nail (grey) and hoof (white) membrane after 120 h of exposure to caffeine, sorbic acid, and testosterone (donor: 20 µg). Box plot with median, minimum, maximum, and the 0.25 and 0.75 quartiles, n = 6.

**Figure 6 pharmaceutics-14-02552-f006:**
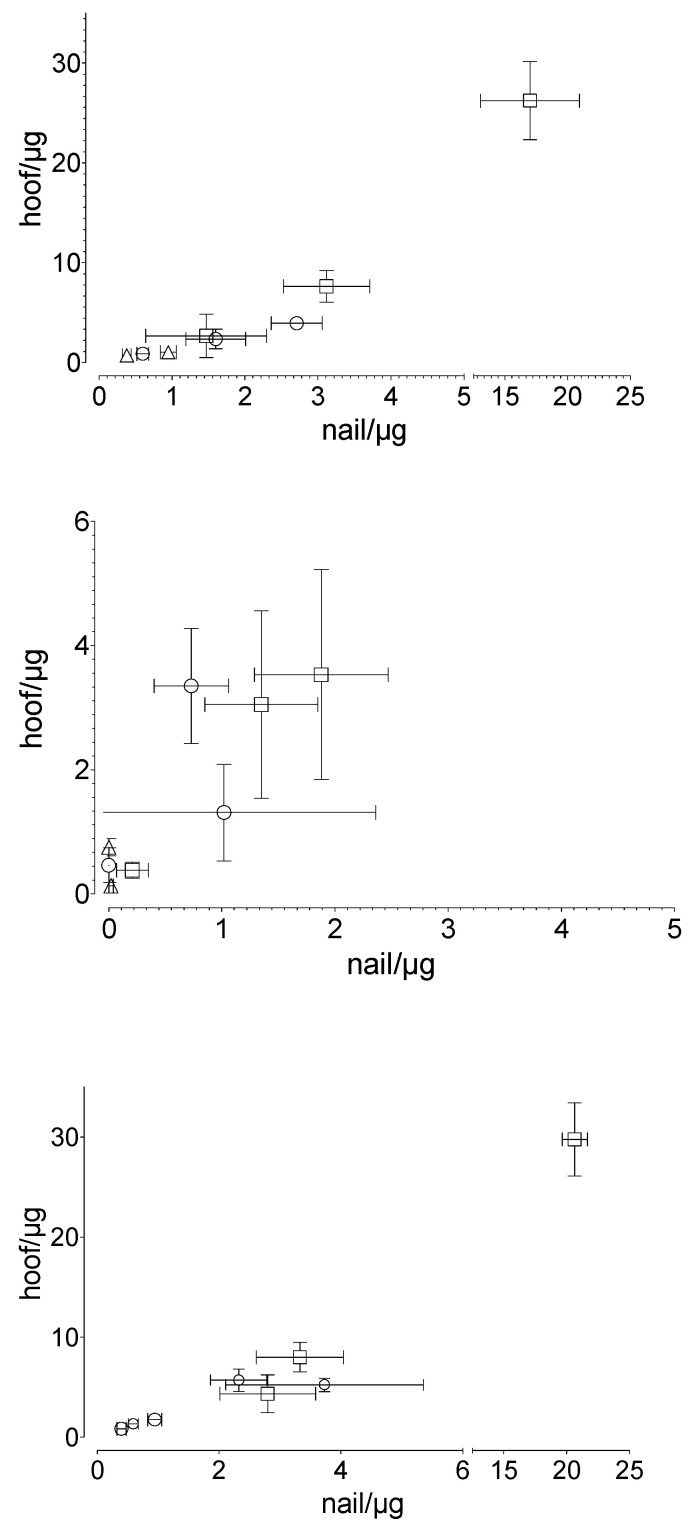
Correlation between human nail and equine hoof membranes in terms of (**top**) the penetrated (R^2^: 0.985), (**middle**) the permeated (R^2^: 0.616), and (**bottom**) the total drug uptake (R^2^: 0.986) drug amount, considering the experimental time and the applied drug amount: ○ caffeine, ☐ sorbic acid, and △ testosterone. Mean value ± standard deviation, *n* = 6.

**Table 1 pharmaceutics-14-02552-t001:** Recovery of caffeine, sorbic acid, and testosterone at the end of the drug uptake studies. Experimental set-up: test period and applied active ingredient. Mean value ± standard deviation, *n* = 6.

Experimental Set-Up	Caffeine/%		Sorbic Acid/%		Testosterone/%	
	Nail	Hoof	Nail	Hoof	Nail	Hoof
24 h/20 µg	103.4 ± 6.0	103.3 ± 7.1	102.8 ± 1.6	97.7 ± 4.1	102.6 ± 0.8	102.5 ± 0.9
24 h/100 µg	98.9 ± 2.0	98.1 ± 0.7	97.1 ± 0.6	91.3 ± 3.9	-	-
120 h/20 µg	102.5 ± 4.1	100.2 ± 4.7	93.5 ± 2.3	82.2 ± 5.7	104.5 ± 2.1	104.9 ± 1.1

**Table 2 pharmaceutics-14-02552-t002:** Physicochemical properties of the studied substances.

	Caffeine	Sorbic Acid	Testosterone
MW (g/mol)	194.19	112.13	288.43
MV (cm^3^) *	133.4 ± 7.0	109.4 ± 3.0	257.0 ± 5.0
PSA (Å^2^) *	58	37	37
logP	−0.13	1.30	3.47
Water solubility (mg/mL)	16	1.6	0.033
pKa		4.76	

MW: molar weight, MV: molar volume, PSA: polar surface area. * The predicted data is generated using the ACD/Labs Percepta Platform—PhysChem Module.

## Data Availability

Not applicable.

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
