# Peer review of "Comparative Ungual Drug Uptake Studies: Equine Hoof Membrane vs. Human Nail Plate"

_pharmaceutics, 2022, doi:10.3390/pharmaceutics14122552_

Round 1

Reviewer 1 Report

Regarding the manuscript (pharmaceutics- 1997384) entitled:

“Comparative ungual drug uptake studies: equine hoof membrane vs. human nail plate”

General comment

The study concentrates on a new permeation device which developed for permeation studies, and the permeation behavior of three model substances on the human nail plate and a model membrane from the horse hoof was investigated. The manuscript, in general, is well written with sufficient data that proved the aim of the study.

Comments

1. Methodology: Drug penetration and permeation studies

Please provide HPLC method for each drug

2. Results: 3.2. Penetration and permeation studies of caffeine, testosterone, and sorbic acid

“A linear correlation was found between the penetrated respectively the permeated amount of the drug in the human nail and that in the equine hoof membrane (Fig. 6).”

- Please provide correlation coefficient and also is it possible the correlation is polynomial not linear?

3. Permeation profiles are common to understand the release pattern with time not just end point. Please provide profile for each drug.

Author Response

Thank you for your engagement and your constructive and helpful comments on our manuscript. Please see the following text for our responses to your comments. We hope that our corrections are in line with your comments and that your concerns have been addressed.

Reviewer 2 Report

GENERAL COMMENT: The methodology proposed in this study is interesting and the model proposed can be very useful to perform ex-vivo transungual penetration experiment for new formulations and substances. Nevertheless, there are some aspects that should be cleared up and completed prior its publication. Please, see specific comments:

SPECIFIC COMMENTS:

Materials and methods – Preparation of the nail bed analogue (ln 92-98): specify supplier for Igepal surfactant.

Materials and methods – Drug penetration and permeation studies (ln 120-140):

- Why Testosterone was not tested at a concentration of 1000µg/mL? because of its poor solubility even with the surfactant and ethanol?

- 100µL of aqueous donor solution was used in all the experiments. It was enough to cover all the diffusional area? Please, specify it.

- The authors pointed out that the remaining donor solutions were washed and analyzed at the end of the experiment (ln 130), what leads to think that maybe the existence of a remaining donor solution suggests that an occlusive condition could take place, leading to the overhydration of the nail. This aspect should be cleared up due to the influence of water content on the penetration.

- HPLC methods for the analysis of the three drugs should be briefly described. Also, it should be stated whether or not these methods were properly validated, with special attention to the LOQ for each substance, since this is a keypoint in this kind of experiments where the absorption is very low.

Stability of sorbic acid with keratin (ln 142-143): When reading the text one cannot explain why the authors perform this experiment. The reason is not clear until line 244. Please, include this explanation under material and methods for clarity.

Results: Figure 2: I miss another water uptake plot for human nail.

Results: Figure 3: there is no permeation of testosterone into through the nail according to the plot a) but the penetration into nail is similar to the other molecules (plot b) this fact is not sufficiently discussed.

Also, I miss the numerical data for each plot for better comparison.

Statistical analysis of the results: I cannot find a statistic study to determine if the differences between the penetration and permeation from both membranes were significant, for each compound. Please, include this approach to justify at the end if the equine hoof is a good model or not in comparison to human nail.

Author Response

(The authors gave the same response as above.)

Reviewer 3 Report

Penetration and permeation studies attempting at developing a reliable model for treatment of human nail diseases is hampered by the fact that none of the models equals the human nail. A membrane cut out from equine or bovine hooves is not identical to human nails as the latter is a complex anatomical structure with differences in the different layers of the nails. Further, the direction of cutting keratin lamellae for these studies has not been paid attention to. It makes a big difference whether the lamellae are cut parallel to the direction of the majority of the keratin fibers or perpendicularly; just look at cut wood how different this looks. Also the human nail consists of layers with a different micromorphology. This has to be kept in mind and discussed.Further, the human nail is the end product of the whole process of nail formation and not the target of most nail diseases except for the rare forms of superficial and endonyx onychomycoses. Even getting a drug through the nail  plate does not mean you can reach the matrix, which is situated under the proximal nail fold although. admittedly, you may reach the nail bed as the main site of fungal nail infection. 

Line 15: "Human nail diseases, mostly caused by fungal infections, are common and complicated to treat." Pls change to "Human nail diseases, mostly caused by fungal infections, are common and difficult to treat." "Complicated" has a slightly different meaning.

Line 24: "... and enables the use different membrane diameters and .."  Insert "of": "...and enables the use of different membrane diameters and .."

Lines 37-38: "..  half of all nail disorders are of infectious origin, 15% are due to inflammatory or metabolic conditions, and 5% are due to malignancies and pigment disturbances." What about the remaining 30%?

Line 39: "... onychomycosis and psoriasis, which require long and systematic therapies." Do you really mean "systematic" treatment or rather "systemic"? I guess rather the latter!

Line 41: "..limited vascular access to the nail bed and .." There is ample nail bed vasculature! Does the nail bed really have a barrier function?

Lines 89-91: "Human nail material, on the other hand, was collected from healthy female volunteers aged between 35 and 60 years, and used without any further treatment. The nails were free from cracks and nail polish. They were hydrated as described above, and punched out with a 6 mm hollow punch." In order to be able to punch out 6 mm discs you need virtually the whole nail. So , were the nails avulsed from the female volunteers?

Results: The recovery of testosterone is over 100%: Does this mean you got out more than you put in?

Author Response

(The authors gave the same response as above.)

Round 2

Reviewer 2 Report

No extra comments. I consider that my questions were adequately answered.